# Stratified analysis of the association between periodontitis and female breast cancer based on age, comorbidities and level of urbanization: A population-based nested case-control study

**Chien-Chih Chen**[1,2]**, Wei-Li Ho**[3]**, Ching-Heng Lin**[4]**, Hsin-Hua Chen**[1,5,6,7,8,9] *

1 Program in Translational Medicine, National Chung-Hsing University, Taichung, Taiwan, 2 Department of Radiation Oncology, Taichung Veterans General Hospital, Taichung, Taiwan, 3 Division of Allergy, Immunology and Rheumatology, Chiayi Branch, Taichung Veterans General Hospital Taichung, Taiwan, 4 Department of Medical Research, Taichung Veterans General Hospital, Taichung, Taiwan, 5 School of Medicine, National Yang-Ming University, Taipei, Taiwan, 6 Institute of Biomedical Science and Rong Hsing Research Center for Translational Medicine, Chung-Hsing University, Taichung, Taiwan, 7 Department of Industrial Engineering and Enterprise Information, Tunghai University, Taichung, Taiwan, 8 Institute of Public Health and Community Medicine Research Center, National Yang-Ming University, Taipei, Taiwan, 9 Division of General Medicine, Department of Internal Medicine, Taichung Veterans General Hospital, Taichung, Taiwan

* shc5555@hotmail.com

## Abstract

### Purpose

To conduct stratified analysis of the association between periodontitis exposure and the risk of female breast cancer based on age, comorbidities and level of urbanization.

### Methods

Using claims data taken from the 1997–2013 Taiwanese National Health Insurance Research Database (NHIRD), we identified 60,756 newly-diagnosed female breast cancer patients during the period 2003–2013 from all beneficiaries. We then randomly selected 243,024 women without breast cancer matching (1:4) for age and the year of the index date during 1997–2013 from a one million representative population acting as the control group. A conditional logistic regression analysis was used to examine the association between peri-odontitis (ICD-9-CM codes 523.3–4) and the risk of breast cancer, shown as an odds ratio (OR) with a 95% confidence interval (CI) after adjustments for the Charlson Comorbidity Index (CCI) and level of urbanization. Subgroup analyses were conducted based on age, CCI and level of urbanization.

### Results

The mean ± standard deviation age was 53 ± 14 years. After adjusting for potential con-founders, the risk of female breast cancer was found to be associated with a history of

**Data Availability Statement:** All relevant data are within the manuscript.

**Funding:** The author(s) received no specific funding for this work.

**Competing interests:** NO authors have competing interests.

periodontitis (OR, 1.12; 95% CI, 1.10–1.14). Such an association was significantly different between patients aged < 65 years (OR, 1.09; 95% CI, 1.06–1.11) and patients aged ≥ 65 years (OR, 1.23; 95% CI, 1.18–1.28; p for interaction <0.001), as well as between patients where the CCI = 0 (OR, 1.17; 95% CI, 1.15–1.20) and patients with CCI > 0 (OR, 0.99; 95% CI, 0.96–1.03; p for interaction <0.001). The highest level of urbanization was also associated with the risk of breast cancer.

## Conclusions

This population-based nested case-control study demonstrated that periodontitis was significantly associated with the risk of female breast cancer and such an association was modified by both age and comorbidities.

## Introduction

Female breast cancer is one of the most common malignancies among women worldwide [1]. In Taiwan, the age-standardized incidence rate of female breast cancer per 100,000 persons was 44.5 and significantly increased during 2000–2006 [2], with the mean hospital treatment cost and length of stay also having increased [3]. Prior studies have shown that the age of menarche, pregnancy, breastfeeding, late menopause, obesity, alcohol use and lack of physical activity were risk factors for breast cancer [4, 5], and a recent study has demonstrated that BRCA mutation carriers experience more severe disease and poorer outcomes [6].

Periodontitis is a chronic oral inflammation condition which is altered by microbiota and the microenvironment [7]. Several reports have shown the association between periodontitis and various chronic inflammatory diseases, including coronary heart disease, stroke, pneumonia, chronic obstructive pulmonary disease, chronic kidney disease, and diabetes mellitus [7–10], which may be due to an altered immune cell function [11–13]. Because immune cell function is affected by periodontitis, this response may theoretically correlate with the development of cancer. Certain studies have demonstrated that periodontitis has also been revealed as having an association with an increased risk of several cancers, including esophageal cancer, head and neck cancer, as well as lung cancer [14–18].

In Taiwan, the prevalence of periodontitis also significantly increased from 11.5% to 19.59% during the period 1997 to 2013 [19]. Chung et al. [20] observed an increased risk of a number of cancers among chronic periodontitis patients, and the adjusted hazard ratio was 1.23 for breast cancer. The available data regarding the association between periodontitis and breast cancer is limited. Soder et al. [21] analyzed 3,273 patients and revealed that severe periodontal disease increased the risk of breast cancer. Shao et al. [22] demonstrated a meta-analysis study and found periodontal disease may be a risk of female breast cancer. However, no population-based study with a large sample size has investigated the relationship. The data from the Taiwanese Health Insurance Research Database (NHIRD) had facilitated nationwide, population-based longitudinal studies. Therefore, in this study we used data taken from the NHIRD to assess the relationship between periodontitis and female breast cancer.

## Materials and methods

### Ethics statement

The study was permitted by the Institutional Review Board (IRB) of Taichung Veterans General Hospital (IRB Number: CE17100B). The requirement for informed consent was waived given that personal information was anonymized.

### Study design

This study was a nationwide, retrospective population-based nested case-control study.

### Data source

**Claim data from the NHIRD in Taiwan.** The study data included the 1997–2013 administrative data from the Taiwanese NHIRD. The National Health Insurance (NHI) program currently covers over 99% of the Taiwanese population. The data found within the NHIRD includes medication prescription history, ambulatory care services, admission services and traditional medical services. Certain personal data and history data, such as body weight, body height, alcohol use, and smoking habits, are not available in the NHIRD. The National Health Research Institute (NHRI) manages the NHIRD and provides the database to researchers for research purposes after anonymization of personal information is assured. The study used several NHIRD datasets, including enrolment files, inpatient and outpatient 1997–2013 claims files and NHI catastrophic illness files.

**Catastrophic illness files.** The NHI registered patients with catastrophic or major diseasese, including cancer and several autoimmune diseases such as systemic lupus erythematosus and rheumatoid arthritis. The NHIRD distributed a package of NHIR catastrophic illness files which include claims of inpatient and outpatient claims and registry details of patients in the catastrophic illness registry. Patients with a catastrophic illness can apply a cartastrophic illness certificate (CIC). The Bureau of NHI (BNHI) validated the accuracy of these catastrophic illness diagnoses in patients who apply CIC by two or more specialists through review of related laboraty data, medical records, pathological data and imaging findings. Patients who meet the diagnositic criteria of a catastrophic illness are issued a CIC and free from co-payment. We utilized the catastrophic illness files in the NHIRD to select newly diagnosed female breast cancer patients and matched patients found during the period 2003 to 2013.

**Longitudinal Health Insurance Database.** In 2000, the NHRI randomly selected and enrolled one million representative individuals from the NHIRD, establishing the Longitudinal Health Insurance Database (LHID2000). Comprehensive information on enrolment and utilization associated this randomly selected cohort are available. We selected a matched non-breast cancer comparison cohort from the population found in the LHID2000. We used LHID2000 claims data from 2003 to 2013 for analysis of the comparison cohort.

### Study subjects

The flowchart of study subjects inclusion was shown in Fig 1.

**Identification of female breast cancer patients from the entire Taiwanese population.** Patients who had a CIC for breast cancer (International Classification of Diseases, Ninth Revision, Clinical Modification [ICD-9-CM] code 174) were considered female breast cancer patients. Using the 1997–2013 NHIRD, we identified 87,738 newly diagnosed female breast cancer patients from 2003 to 2013. The first date of CIC application was defined as the index date.

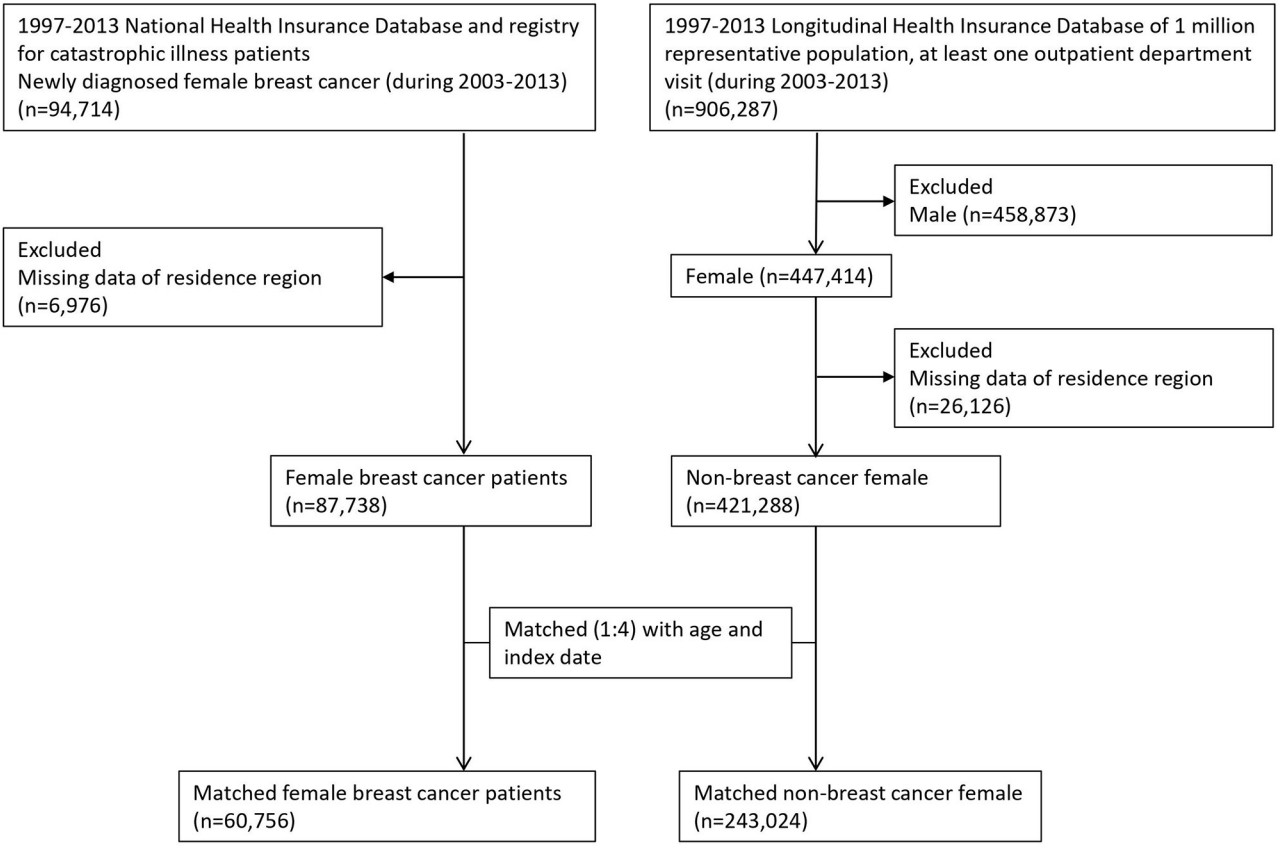

**Fig 1. Flow chart of the study design.**

**Selection of female breast cancer patients and age-matched non-breast cancer female controls.** Using claim data from the 1997–2013 LHID2000, we identified 421,288 females who never had a diagnosis of breast cancer (ICD-9-CM code 174). We matched female breast cancer patients with females without breast cancer at a 1:4 ratio for age and year of the index date and finally included 60,756 female breast cancer patients and 243,.24 matched controls in the study.

## Definition of periodontitis exposure

In Taiwan, beneficiaries of NHI were allowed to receive dental scaling no more than twice per year. Dentists may code gingival or periodontal disease for ambulatory visits of patient who received regular dental scaling. Therefore, in the study, only patients who had at least one outpatient visit before the index date with a diagnosis of acute or chronic periodontitis (ICD-9-CM codes 523.3–4) and concurrently received antibiotic treatment, or periodontal treatment, or scaling more than twice a year by dentists, were identified as patients with periodontitis.

## Potential confounders

Potential confounders included level of urbanization and CCI. The CCI, as adapted by Deyo et al. [23], was used to represent the general level of comorbidity medical conditions. The

presence of comorbidity was defined as a patient having at least three ambulatory visits or one inpatient visit with a corresponding ICD-9-CM code within 1 year prior to the index date. The level of urbanization of patient's residence was categorized into four clusters according to density of population (people/km$^2$), ratio of elderly individuals aged $> 65$ years, population ratio of subjects with educational levels of college or above, number of physicians/$10^5$ subjects, and population ratio of agricultural workers [24]. The variable level of urbanization is an ecological level variable since it describes properties of the region.

## Proxy measures for severity of periodontitis to assess dose-response relationship with female breast cancer risk

In order to measure the relationship between the severity of periodontitis and the risk of female breast cancer, the period between the last periodontitis visit date to breast cancer diagnosis date (i.e., 0–3 months, 3–6 months, 6–1 months, 1–3 years, $> 3$ years), the number of visits required for periodontitis according to the 25th, 50th and 75th centiles, and the cumulative cost of periodontitis-related visits according to the 25th, 50th and 75th centiles were incorporated into the analysis.

## Sensitivity analysis

Sensitivity analyses were conducted by using various definitions of gingival and periodontal disease based on ICD-9-CM codes (i.e., gingival and periodontal disease: ICD-9-CM codes 523, chronic periodontitis: ICD-9-CM code 523.4, periodontal disease: ICD-9-CM codes 523–3–5) to test whether the results remained robust.

## Subgroup analysis

We conducted subgroup analyses based on age (i.e., $<65$ years, $\geq 65$ years), CCI (i.e., 0, $\geq 1$) and level of urbanization to test whether the results remained robust in various subgroups and to examine whether or not these variables have modification effects.

## Statistical analysis

We compared continuous variables including age and CCI using the Student's $t$-test and the categorical variables including variables related to periodontitis history and level of urbanization by using the Pearson's chi-square test between cases and controls. A multivariable conditional logistic regression with maximum likelihood estimation was used to examine the association between periodontitis exposure and the risk of breast cancer development after adjustment for potential confounders shown as adjusted odds ratio (aOR) with 95% confidence interval (CI). Potential confounders included CCI and the level of urbanization (i.e., Leve1 1 to 4) [24]. The significance of modification effect by each covariate on periodontitis exposure-associated female breast cancer risk was examined by calculating the $P$-value of the coefficient associate with the product of each indicator of the covariate and the indicator of periodontitis using the Wald test.

   These statistical analyses were performed using SAS statistical software, version 9.3 (SAS Institute, Inc., Cary, NC, USA). A p value less than 0.05 was considered statistically significant.

## Results

Table 1 demonstrated the demographic and clinical data of the matched female breast cancer patients and the non-breast cancer female controls. The mean ± SD age was 53±14 years in both groups. Female breast cancer patients had a higher proportion of having a history of

**Table 1. Demographic and clinical data of patients with breast cancer and non-breast cancer controls.**

| Variable | Female non-breast cancer patients (n = 243,024) | Female breast cancer patients (n = 60,756) | P-value |
|---|---|---|---|
| **Age**, years (mean ± SD) | 53±14 | 53±14 | 1.000 |
| **Gingival and periodontal disease definitions by ICD-9-CM** | | | |
| Gingival and periodontal diseases (ICD-9-CM: 523) | 106,081 (43.7) | 28,358 (46.7) | <0.001 |
| Acute or chronic periodontitis (ICD-9-CM: 523.3–4) | 69,604 (28.6) | 19,018 (31.3) | <0.001 |
| Chronic periodontitis (ICD-9-CM: 523.4) | 18,917 (7.8) | 5,343 (8.8) | <0.001 |
| Periodontitis (ICD-9-CM: 523.3–5) | 94,477 (38.9) | 25,357 (41.7) | <0.001 |
| **Last periodontitis visit date to breast cancer diagnosis date** | | | <0.001 |
| 0–3 months | 4,065 (1.7) | 1,466 (2.4) | |
| 3–6 months | 4,080 (1.7) | 1,201 (2.0) | |
| 6–12 months | 7,023 (2.9) | 1,884 (3.1) | |
| 1–3 years | 19,505 (8.0) | 5,162 (8.5) | |
| >3 years | 34,931 (14.4) | 9,305 (15.3) | |
| **Number of visits for periodontitis** | | | <0.001 |
| Q1, Q2 (1) | 51,369 (21.1) | 13,917 (22.9) | |
| Q3 (2) | 10,824 (4.5) | 2,985 (4.9) | |
| Q4 (>2) | 7,411 (3.0) | 2,116 (3.5) | |
| **Cumulative cost of periodontitis-related visits (US dollars)** | | | <0.001 |
| Q1 (0–13) | 17,902 (7.4) | 4,662 (7.7) | |
| Q2 (14–16) | 18,314 (7.5) | 5,269 (8.7) | |
| Q3 (17–43) | 16,964 (7.0) | 4,454 (7.3) | |
| Q4 (>43) | 16,424 (6.8) | 4,633 (7.6) | |
| **Level of urbanization** | | | <0.001 |
| 3 (least urbanization) | 16,612 (6.8) | 3,085 (5.1) | |
| 2 | 35,724 (14.7) | 7,680 (12.6) | |
| 1 | 108,933 (44.8) | 26,900 (44.3) | |
| 0 (most urbanization) | 81,755 (33.6) | 23,091 (38.0) | |
| **CCI** | 0.4±1.0 | 1.2±2.3 | <0.001 |
| **CCI group** | | | |
| 0 | 196,320 (80.8) | 41,007 (67.5) | <0.001 |
| ≥1 | 46,704 (19.2) | 19,749 (32.5) | |

Data are shown as number (percentage) unless specified otherwise.

Abbreviations: ICD-9-CM, International Classification of Diseases, Ninth Revision, Clinical Modification; Q, quartile; CCI, Charlson comorbidity index.

periodontitis than the control group. Female breast cancer patients had a higher proportion of living in the more urbanized regions than controls. The period between the last periodontitis visit date to the index date was shorter in the female breast cancer patients than in the matched controls. The number of periodontitis-related visits and the cumulative cost of periodontitis related visits were higher in female breast cancer patients than in controls. Female breast cancer patients also had a higher CCI than controls.

Table 2 revealed crude and adjusted OR with 95% CI for the associations of female breast cancer with periodontitis and other covariates. After adjusting for potential confounders, periodontitis was significantly associated with the risk of female breast cancer (aOR, 1.12; 95% CI, 1.09–1.14). CCI ≥1 and higher level of urbanization were also risk factors for female breast cancer.

**Table 2. Crude and adjusted OR with 95% CI for the association between variables and female breast cancer using conditional logistic regression analyses.**

| Variable | Univariable analysis | Multivariable analysis* |
|---|---|---|
| **Periodontitis** | 1.14 (1.12–1.17) | 1.12 (1.09–1.14) |
| **CCI group** | | |
| 0 | 1.00 (Reference) | 1.00 (Reference) |
| ≥1 | 2.31 (2.26–2.36) | 2.32 (2.27–2.37) |
| **Level of urbanization** | | |
| 3 (least urbanization) | 1.00 (Reference) | 1.00 (Reference) |
| 2 | 1.16 (1.11–1.21) | 1.17 (1.127–1.22) |
| 1 | 1.34 (1.28–1.39) | 1.36 (1.30–1.41) |
| 0 (most urbanization) | 1.53 (1.47–1.60) | 1.57 (1.50–1.63) |

*Adjusting for all variables.

Abbreviations: OR, odds ratio; CI, confidence interval; CCI, Charlson comorbidity index

As shown in Table 3, the association between periodontitis and breast cancer was consistent using various definitions of periodontitis, implying that acute and/or chronic periodontitis was associated with breast cancer.

Table 4 shows the OR with 95% CI associated with variables for the risk of female breast cancer. After adjusting for age and, the risk of female breast cancer was associated with a history of periodontitis (OR, 1.12; 95% CI, 1.10–1.14). A short interval period between the last periodontitis visit date to the breast cancer diagnosis date (OR, 1.46; 95% CI, 1.37–1.55), more periodontitis visits (OR, 1.17; 95% CI, 1.11–1.23), a higher cumulated periodontitis-related cost (OR, 1.15; 95% CI, 1.11–1.19), and higher urbanization level were all associated with the risk of breast cancer.

Table 5 shows that the association was significantly different between patients < 65 years of age (OR, 1.09; 95% CI, 1.06–1.11) and patients ≥ 65 years of age (OR, 1.23; 95% CI, 1.18–1.28; p for interaction <0.001), as well as between patients without the CCI (OR, 1.17; 95% CI, 1.15–1.20) and patients with the CCI (OR, 0.99; 95% CI, 0.96–1.03; p for interaction <0.001). The supplementary figures showed the partial effect plots to summarize the statistical models.

## Discussion

This is a nationwide, population-based nested case-control study which assessed the association between a history of periodontitis and the risk of female breast cancer development in

**Table 3. Sensitivity analysis for the association of female breast cancer with gingival and periodontal disease using various definitions based on ICD-9-CM coder.**

| Gingival and periodontal disease definition by ICD-9-CM code | Univariable | Multivariable* |
|---|---|---|
| Gingival and periodontal diseases (ICD-9-CM code 523) | 1.14 (1.12–1.16) | 1.11 (1.09–1.13) |
| Periodontitis (ICD9-CM codes 523.3–4) | 1.14 (1.12–1.17) | 1.12 (1.10–1.14) |
| Chronic periodontitis (ICD-9-CM code 523.4) | 1.15 (1.11–1.18) | 1.11 (1.07–1.15) |
| Periodontal disease (ICD-9-CM codes 523.3–5) | 1.13 (1.11–1.16) | 1.10 (1.08–1.12) |

Statistical analyses were conducted using a conditional logistic regression model.

*Adjusting for Charlson comorbidity index group (0, ≥1) and level of urbanization.

Abbreviations: ICD-9-CM, International Classification of Diseases, Ninth Revision, Clinical Modification.

**Table 4. Crude and adjusted odds ratios with 95% confidence intervals for the association between history of periodontitis and female breast cancer by conditional logistic regression analyses.**

| History of periodontitis | Univariable analysis | Multivariable analysis* |
|---|---|---|
| **Interval between the last periodontitis visit and the index date** | | |
| No periodontitis | 1.00 (Reference) | 1.00 (Reference) |
| 0–3 months | 1.51 (1.42–1.60) | 1.45 (1.36–1.54) |
| 3–6 months | 1.23 (1.15–1.31) | 1.19 (1.12–1.27) |
| 6–12 months | 1.12 (1.07–1.18) | 1.09 (1.04–1.15) |
| 1–3 years | 1.11 (1.07–1.14) | 1.08 (1.04–1.11) |
| >3 years | 1.11 (1.09–1.14) | 1.10 (1.07–1.12) |
| **Number of visits for periodontitis** | | |
| No periodontitis | 1.00 (Reference) | 1.00 (Reference) |
| Q1, Q2 (1) | 1.13 (1.11–1.16) | 1.11 (1.09–1.13) |
| Q3 (2) | 1.15 (1.11–1.20) | 1.12 (1.07–1.17) |
| Q4 (>2) | 1.20 (1.14–1.26) | 1.17 (1.11–1.23) |
| **Cumulative cost of periodontitis-related visits (US dollars)** | | |
| No periodontitis | 1.00 (Reference) | 1.00 (Reference) |
| Q1 (0–13) | 1.09 (1.05–1.12) | 1.07 (1.03–1.11) |
| Q2 (14–16) | 1.21 (1.17–1.25) | 1.19 (1.15–1.23) |
| Q3 (17–43) | 1.10 (1.06–1.14) | 1.07 (1.03–1.10) |
| Q4 (>43) | 1.18 (1.14–1.22) | 1.15 (1.11–1.19) |

*Adjusting for Charlson comorbidity index group (0, ≥1) and level of urbanization.

Abbreviations: Q, quartile.

Asia, and we have demonstrated a significant association of female breast cancer with periodontitis. The results remained robust in patients aged <65 years or ≥65 years and patients with a CCI = 0. Consistent with our finding, Soder et al. [21] selected 3,273 patients aged between 30–40 years and found that chronic periodontitis indicated by missing molars seemed to be associated with breast cancer (odds ratio 2.36). Of note, a meta-analysis study [22] found

**Table 5. Stratified analysis of the association between periodontitis and female breast cancer based on age, comorbidities and level of urbanization: A population-based nested case-control study*.**

| Subgroup | OR (95% CI) | p value | p for interaction |
|---|---|---|---|
| **Age group** | | | <0.001 |
| Age<65 | 1.09 (1.06–1.11) | <0.001 | |
| Age≥65 | 1.23 (1.18–1.28) | <0.001 | |
| **CCI group** | | | <0.001 |
| 0 | 1.17 (1.15–1.20) | <0.001 | |
| ≥1 | 0.99 (0.96–1.03) | 0.652 | |
| **Level of urbanization group** | | | 0.727 |
| 0 (least urbanization) | 1.12 (1.03–1.22) | 0.009 | |
| 1 | 1.15 (1.09–1.21) | <0.001 | |
| 2 | 1.11 (1.08–1.14) | <0.001 | |
| 3 (most urbanization) | 1.09 (1.06–1.13) | <0.001 | |

*Adjusted variables included age, CCI and level of urbanization with the exclusion of the subgroup variable.

Abbreviations: CCI, Charlson comorbidity index; OR, odds ratio; CI, confidence interval.

a periodontal disease significantly increased breast cancer risk (RR, 1.22; 95% CI, 1.06–1.40). However, the risk of breast cancer development was not significantly increased among patients with periodontal disease and a history of periodontal therapy. Therefore, the author concluded a potential association between periodontitis and breast cancer. Freudenheim et al. [25] analyzed a cohort of 73,737 postmenopausal women and also revealed that periodontitis was associated with the risk of postmenopausal breast cancer (HR, 1.14; 95% CI, 1.03–1.26). On the contrary, Farhat et al. [26] revealed no association between periodontal disease and overall breast cancer risk (HR, 1.02; 95% CI, 0.94–1.10). However, in this study, no confounder was analyzed. In contrast, we analyzed the association between periodontitis and breast cancer adjusting for potential confounds, and we also found the important modification effects by age and comorbidities.

Another finding of our study was that the magnitude of the association between periodontitis and female breast cancer is strongest when the lag time of the last periodontitis-related visit was <3 months. However, it is possible that occult breast cancer might have occurred before the last periodontitis-related visit when the interval was less than 3 months. Therefore, we cannot exclude the possibility of reverse causality. Further studies are warranted to assess whether breast cancer is a risk factor for periodontitis. Additionally, we also discovered a greater risk of breast cancer in the patients who had more severe periodontitis, suggesting a dose-response relationship between periodontitis and female breast cancer. Consistent with this finding, Soder et al. [21] revealed that breast cancer was associated with severe periodontal disease (periodontal disease and/or missing molars), but not mild periodontal disease (periodontal disease without missing molars).

Another important finding was that the association between periodontitis and female breast cancer was significantly different on the basis of age and CCI. In subgroup analyses, the association between periodontitis and female breast cancer was greater in elderly patients 65 years or older than in younger patients. The reason for this may be that elderly patients may have had a longer history of periodontitis, reflecting a longer exposure to bacteria and greater inflammation, leading to more severe periodontitis.

We also found that periodontitis exposure was not associated female breast cancer in patients with CCI≥1. Cao et al. [27] revealed that overall survival and progression-free survival of breast cancer were impacted by d CCI, implying that some comorbidities may play a mechanistic role in the development and progression of female breast cancer. Therefore, the presence of comorbidities may be a competitive risk factor for breast cancer in patients with periodontitis, leading a negative association between periodontitis and female breast cancer in patients with comorbidities. Another explanation was that patients with comorbidities had a higher chance of breast cancer surveillance than those without comorbidities, leading to a detection bias.

Whether periodontitis may induce breast cancer or not still remains unclear. Oral cavity bacteria may play a role in the potential mechanism. Previous studies [28–30] have found that breast duct tissue was exposed to various bacteria, with bacteria also being found in breast tumors. The long-term bacteria stimulation and inflammation seems to lead to cancer formation. Periodontitis is a chronic inflammation condition and its associated systemic inflammation status may play a role as another mechanism. Certain studies [31–35] have revealed that periodontal disease increased systemic inflammation markers, including C-reactive protein (CRP), cytokines and chemokines. Chan et al. [31] reviewed the meta-analysis of several studies and found that circulating CRP, a low grade inflammation marker, was associated breast cancer development. Noack et al. [33] showed CRP levels increased in periodontitis patients. Hayashi et al. [34] demonstrated oral pathogens can induce and maintain a chronic state of inflammation at sites distant from oral infection. Elinav et al. [35] showed that microbials

played an important role in inflammation-induced cancer and also affected cancer development. These inflammation markers had an impact on carcinogenesis and could explain the association between periodontitis and breast cancer.

We also found that a higher level of urbanization was associated with a greater risk of female breast cancer. This finding may be explained by differential dietary patterns and life stress levels among patients living in areas with various levels of urbanization. Zhang et al. [36] and Fei et al. [37] found that breast cancer shows an urban-rural disparity. Previous studies [38–40] established the association between breast cancer and socioeconomic status, and socioeconomic factors included diet, alcohol consumption, physical exercise, etc.

Some limitations were noted in this study. First, some confounders, including menarche, pregnancy, breastfeeding, menopause, body weight, body height, alcohol use, and smoking, were not made available in this study, and all these factors can affect the risk of breast cancer. Second, the severity of periodontitis may affect one's inflammation status, which could lead to breast cancer. Another limitation is that some variables are continuous variables, but we stratified these variables as categorical variables. In this way, we can find out high-risk populations easier. In order to overcome these limitations, we enrolled large numbers of matched patients and stratified patients by their number of visits for periodontitis treatment and the cumulative cost of their periodontitis-related visits. A higher number of visits for periodontitis treatment, along with a higher cumulative cost for periodontitis-related visits, resulted in patients being considered to have more active and severe periodontitis. Third, although the regular audit by the BNHI had improved coding accuracy, the accuracy of diagnoses based on claims data is still an issue of concern. However, the non-differential misclassification bias related to periodontitis diagnosis always drives the bias towards the null. Therefore, the magnitude of the association between periodontitis and breast cancer could only be underestimated.

In conclusion, this population-based nested case-control study revealed periodontitis exposure is significantly associated with the risk of breast cancer. We also found this association is modified by both age and comorbidities. Further studies are warranted to clarify the mechanisms.

## Supporting information

**S1 Fig. Partial effect plots to show the interaction between age and periodontitis on the probability of female breast cancer.**
(TIF)

**S2 Fig. Partial effect plots to show the interaction between age group and periodontitis on the probability of female breast cancer.**
(TIF)

**S3 Fig. Partial effect plots to show the interaction between CCI and periodontitis on the probability of female breast cancer.**
(TIF)

**S4 Fig. Partial effect plots to show the interaction between CCI group and periodontitis on the probability of female breast cancer.**
(TIF)

## Acknowledgments

This study was supported by Taichung Veterans General Hospital.

## Author Contributions

**Conceptualization:** Ching-Heng Lin.

**Data curation:** Wei-Li Ho, Ching-Heng Lin.

**Supervision:** Hsin-Hua Chen.

**Writing – original draft:** Chien-Chih Chen.

**Writing – review & editing:** Hsin-Hua Chen.

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
