## [Decision Letter · Decision Letter 0]

28 Apr 2022

PONE-D-22-02756Effect Modification by Age and Comorbidities on the Association Between Periodontitis and Female Breast Cancer: A Population-based Case-control StudyPLOS ONE

Dear Dr. Chen,

Thank you for submitting your manuscript to PLOS ONE. After careful consideration, we feel that it has merit but does not fully meet PLOS ONE’s publication criteria as it currently stands. Therefore, we invite you to submit a revised version of the manuscript that addresses the points raised during the review process. There are some issues that have been raised from the reviewers regarding the statistical analyses and the structure and content of the discussion that should be addressed.

We look forward to receiving your revised manuscript.

Kind regards,

Antonis Valachis

Academic Editor

PLOS ONE

Journal Requirements:

“NO authors have competing interests.”

Reviewers' comments:

Reviewer's Responses to Questions

**Comments to the Author**

1. Is the manuscript technically sound, and do the data support the conclusions?

Reviewer #1: Yes

Reviewer #2: Partly

2. Has the statistical analysis been performed appropriately and rigorously? 

Reviewer #1: Yes

Reviewer #2: No

3. Have the authors made all data underlying the findings in their manuscript fully available?

Reviewer #1: Yes

Reviewer #2: No

4. Is the manuscript presented in an intelligible fashion and written in standard English?

Reviewer #1: No

Reviewer #2: Yes

5. Review Comments to the Author

Reviewer #1: Introduction

Page 4. Did the prevalence of periodontitis increase because of changes in definitions? This has often been the case in other countries.

Page 4. There are many other cohort studies on periodontitis and breast cancer that are not referenced (e.g. Farhat et al CEBP 2021, Jia et al CEBP 2020, and earlier meta-analysis of 11 studies: Shao et al 2018). Also, the Sfreddo et al study referenced is not a cohort study. Please edit the last paragraph of the introduction to reflect this. The discussion (first paragraph) also needs to be revised since the association between periodontal disease and breast cancer risk has previously been reported in Taiwan using the same dataset (PMID: 26280747); please explain the differences/overlap between the two studies in the discussion section.

Methods

This is not a traditional case-control study; based on the selection of all breast cancer cases during a 10-year period and selection of controls from a large cohort database for comparison, this would be better described as a retrospective population-based cohort study.

It appears that all incident breast cancer cases identified through the NHIRI between 2003 and 2013 were included in this analysis. It is unclear if the breast cancer cases that are not in the NHRI database would have detailed data on claims for periodontal disease or other co-morbidities and if these were also included. A figure would help between understand the selection process including exclusions that were made and available data available for each step.

Statistical analysis – please provide details on how the tests for interactions were conducted.

Would it be possible to differentiate different types of breast cancers? E.g. invasive vs non-invasive?

Also, given the strong association between urbanization and breast cancer, it would be useful to examine whether the periodontal disease and breast cancer is modified by this variable (this could be added to Table 5).

Discussion

The discussion could be improved. Prior results from existing studies on this topic are not provided (e.g., range of RRs to compare magnitude of association, quality of periodontal disease measurement). It would also be helpful to point out some limitations in prior studies, and what this study adds to the literature (given that there are over 10 papers on this topic). A better synthesis of the literature is needed.

Typos (tables and “CRT” in discussion) and English grammar need to be corrected throughout.

Reviewer #2: This study explored the effect of periodontitis on female breast cancer using a population based case control study. The paper is reasonably well written though I have some comments which the authors might consider to improve their study.

Introduction

Page 4 final paragraph – "small sample size" is somewhat relative – I would state the sample sizes of these studies

Page 5: "...while Sfreddo et al. [21] found there was a significant association between periodontitis and breast cancer" - Presumably this is a positive association? Please clarify in the text

Materials and methods

Page 7 "Potential confounders included level of urbanization". Presumably this is a city / regional level variable in the dataset?

"Sensitivity analyses were conducted based upon age (<65 years, ≥65 years) and CCI" – further elaboration required.

"We tested the differences in the continuous variables through use of the Student’s t-test and the categorical variables by using the Pearson’s χ2 test" What is the motivation for this and define the specific variables?

Logistic regression model – presumably maximum likelihood was used to estimate the model parameters, but it would be good to state this explicitly.

There is a strong case for modelling the dose response relationship as a continuous response. Arbitrarily creating categories is statistically inefficient and at worst can be misleading. I would advise modelling each explanatory variable of interest for the dose response as a continuous term. This can easily be achieved in SAS using a spline. A good choice would be a restricted cubic spline with the number of knots selected based on prior knowledge or beliefs (3-5 knots is often sensible – see Harrell F. (2015) Regression Modelling Strategies). The dose response adjusted for covariates can be presented as a smooth curve with confidence intervals. Odds ratios can still be obtained from this curve between specific points of interest. You might also consider modelling the CCI this way. Even if this approach is not adopted, the authors might consider mentioning this as a limitation.

Details of the interaction term (later referred to in subsequent sections) in the logistic regression models are missing. This is critical since the primary interest (according to the title of the paper) is effect modification necessitating interaction terms.

Perhaps the authors might consider making dataset and SAS code used available as SI

Results

Page 9 "After adjusting for age and index date"…this implies that age and index date were covariates in the regression model – this does not appear to be the case according to the statistical analysis section as it stands.

It would be nice to have some supportive figures (e.g. partial effect plots) summarizing the statistical models.

Discussion

First paragraph talks about several studies one by one – it would be preferable to integrate this as continuous prose and convey the message that your work is supportive of these previous studies.

"In this study, one important finding is that the risk of breast cancer is modified by both age and comorbidities" - this statement is redundant since it is a repetition of the first sentence of the paragraph.

Table 3 – It seems patient is the statistical unit as opposed to an adjustment variable as currently indicated at the foot of the table. Also urban should be "urbanization".

6. PLOS authors have the option to publish the peer review history of their article (what does this mean?). If published, this will include your full peer review and any attached files.

Reviewer #1: No

Reviewer #2: No

---

## [Author Response · Author response to Decision Letter 0]

9 Jun 2022

Reviewer #1: Introduction

1.Page 4. Did the prevalence of periodontitis increase because of changes in definitions? This has often been the case in other countries.

Response: Thanks for your comment! Yes, the prevalence of periodontitis varied with changes in definitions based on ICD-9-CM codes (Table 1). Therefore, we conducted a sensitivity analysis for the association between periodontitis and breast cancer using various definitions of periodontitis and found consistent results with ORs from 1.10 to 1.12. Also, the non-differential misclassification bias related to periodontitis diagnosis always drives the bias towards the null. Therefore, the magnitude of the association between periodontitis and breast cancer could only be underestimated. We added a description regarding the influence of periodontitis definition on its association with breast cancer in the last paragraph of the Discussion section: ‘Third, although the Bureau of NHI audit had improved coding accuracy, the accuracy of diagnoses based on claims data is still an issue of concern. However, the non-differential misclassification bias related to periodontitis diagnosis always drives the bias towards the null. Therefore, the magnitude of the association between periodontitis and breast cancer could only be underestimated.’

2.Page 4. There are many other cohort studies on periodontitis and breast cancer that are not referenced (e.g. Farhat et al CEBP 2021, Jia et al CEBP 2020, and earlier meta-analysis of 11 studies: Shao et al 2018). Also, the Sfreddo et al study referenced is not a cohort study. Please edit the last paragraph of the introduction to reflect this. The discussion (first paragraph) also needs to be revised since the association between periodontal disease and breast cancer risk has previously been reported in Taiwan using the same dataset (PMID: 26280747); please explain the differences/overlap between the two studies in the discussion section.

Response: Shao et al [21] demonstrated a meta-analysis study and found periodontal disease may be a risk of female breast cancer. (page 4)

A meta-analysis study [21] found a potential association between periodontitis and breast cancer. (page 11)

In contrast, we analyzed the association between periodontitis and breast cancer in females, and we also found the important modification effects by age and comorbidities. (page 11)

3.Methods

This is not a traditional case-control study; based on the selection of all breast cancer cases during a 10-year period and selection of controls from a large cohort database for comparison, this would be better described as a retrospective population-based cohort study.

Response: Because we retrospectively selected the population of interested based on the outcome (i.e., incident female breast cancer patients as cases and female individuals without breast cancer as controls) and identified the history of periodontitis exposure before the index date, this study was a retrospective population-based case-control study. (page 5)

4.It appears that all incident breast cancer cases identified through the NHIRI between 2003 and 2013 were included in this analysis. It is unclear if the breast cancer cases that are not in the NHRI database would have detailed data on claims for periodontal disease or other co-morbidities and if these were also included. A figure would help between understand the selection process including exclusions that were made and available data available for each step.

Response: Taiwan's National Health Insurance (NHI) began in March 1, 1995 and covered 99.9% of Taiwan's residents by 2014. If the female breast cancer cases were not included in the NHRI database, data on claims for periodontal disease or other comorbidities should be unavailable. We added a figure to show the process of study subjects inclusion.

5.Statistical analysis – please provide details on how the tests for interactions were conducted.

Response: We had provided details on how to test the interaction effects in the statistical method subsection in the Methods section: ‘The significance of modification effect by each covariate on periodontitis exposure-associated female breast cancer risk was examined by calculating the P-value of the coefficient associate with the product of each indicator of the covariate and the indicator of periodontitis using the Wald test.’ (page 9)

6.Would it be possible to differentiate different types of breast cancers? E.g. invasive vs non-invasive?

Response: Female breast cancer patients were defined as those having a Catastrophic Illness Certificate (CIC) for breast cancer [International Classification of Diseases, Ninth Revision, Clinical Modification (ICD-9-CM) code 174]. This criteria only included the invasive breast cancer patients.

7.Also, given the strong association between urbanization and breast cancer, it would be useful to examine whether the periodontal disease and breast cancer is modified by this variable (this could be added to Table 5).

Response: We have examined the modification effect of urbanization on the association between periodontitis and female breast cancer and added the results to Table 5. 

8.Discussion

The discussion could be improved. Prior results from existing studies on this topic are not provided (e.g., range of RRs to compare magnitude of association, quality of periodontal disease measurement). It would also be helpful to point out some limitations in prior studies, and what this study adds to the literature (given that there are over 10 papers on this topic). A better synthesis of the literature is needed.

Response: Soder et al. [20] selected 3,273 patients aged between 30-40 years and found that chronic periodontitis indicated by missing molars seemed to be associated with breast cancer (odds ratio 2.36). Of note, a meta-analysis study [21] found a periodontal disease significantly increased breast cancer risk (RR = 1.22, 95% CI: 1.06-1.40). However, the risk of breast cancer development was not significantly increased among patients with periodontal disease and a history of periodontal therapy. Therefore, the author concluded a potential association between periodontitis and breast cancer. Freudenheim et al. [24] analyzed a cohort of 73,737 postmenopausal women and also revealed that periodontitis was associated with the risk of postmenopausal breast cancer (HR 1.14, 95% CI: 1.03-1.26). On the contrary, Farhat et al. [25] revealed no association between periodontal disease and overall breast cancer risk (HR 1.02, 95% CI: 0.94-1.10). However, in this study, no confounder was analyzed. In contrast, we analyzed the association between periodontitis and breast cancer adjusting for potential confounds, and we also found the important modification effects by age and comorbidities. (page 11)

9.Typos (tables and “CRT” in discussion) and English grammar need to be corrected throughout.

Response: We corrected it. (page 13)

Reviewer #2: This study explored the effect of periodontitis on female breast cancer using a population based case control study. The paper is reasonably well written though I have some comments which the authors might consider to improve their study.

10.Introduction

Page 4 final paragraph – "small sample size" is somewhat relative – I would state the sample sizes of these studies

Response: Soder et al. [20] analyzed 3,273 patients and revealed that severe periodontal disease increased the risk of breast cancer. (page 4)

11.Page 5: "...while Sfreddo et al. [21] found there was a significant association between periodontitis and breast cancer" - Presumably this is a positive association? Please clarify in the text

Response: We clarified the results of the meta-analysis study by Sfreddo et al. [21]: ‘ Of note, a meta-analysis study [21] found a periodontal disease significantly increased breast cancer risk. However, the risk of breast cancer development was not significantly increased among patients with periodontal disease and a history of periodontal therapy. Therefore, the author concluded a potential association between periodontitis and breast cancer.’ (page 11)

12.Materials and methods

Page 7 "Potential confounders included level of urbanization". Presumably this is a city / regional level variable in the dataset?

Response: Thanks for your comment. The variable of level of urbanization is an individual level variable based on individual’s residence region in the dataset.

13."Sensitivity analyses were conducted based upon age (<65 years, ≥65 years) and CCI" – further elaboration required.

Response: We are sorry that this is a mistake. “Sensitivity” analyses should be replaced with “Subgroup” analyses. Conducted subgroup analyses and examined the modification effects by age group and CCI group on the association between periodontitis exposure and the risk of female breast cancer. We also added a description regarding the statistical method to assess the modification effect in the statistical method subsection in the Methods section: ‘The significance of modification effect by each covariate on periodontitis exposure-associated female breast cancer risk was examined by calculating the P-value of the coefficient associate with the product of each indicator of the covariate and the indicator of periodontitis using the Wald test.’(page 9)

14."We tested the differences in the continuous variables through use of the Student’s t-test and the categorical variables by using the Pearson’s χ2 test" What is the motivation for this and define the specific variables?

Response: We intended to compare the continuous variables and categorical variables between cases and controls. To clarify our motivation of this and define the specific variables, we revised our description as: ‘We compared continuous variables including age and CCI using the Student’s t-test and the categorical variables including variables related to periodontitis history and level of urbanization by using the Pearson’s chi-square test between cases and controls.’

15.Logistic regression model – presumably maximum likelihood was used to estimate the model parameters, but it would be good to state this explicitly.

Response: We revised the description regarding statistical analysis in the statistical analysis subsection in the Methods section as:’ A multivariable conditional logistic regression with maximum likelihood estimation was used to examine the association between periodontitis exposure and the risk of breast cancer development after adjustment for potential confounders shown as adjusted odds ratio (aOR) with 95% confidence interval (CI).’ 

16.There is a strong case for modelling the dose response relationship as a continuous response. Arbitrarily creating categories is statistically inefficient and at worst can be misleading. I would advise modelling each explanatory variable of interest for the dose response as a continuous term. This can easily be achieved in SAS using a spline. A good choice would be a restricted cubic spline with the number of knots selected based on prior knowledge or beliefs (3-5 knots is often sensible – see Harrell F. (2015) Regression Modelling Strategies). The dose response adjusted for covariates can be presented as a smooth curve with confidence intervals. Odds ratios can still be obtained from this curve between specific points of interest. You might also consider modelling the CCI this way. Even if this approach is not adopted, the authors might consider mentioning this as a limitation.

Response: We have modelled the dose response relationship of age and CCI for the interaction with periodontitis as a continuous response and showed the results in supplementary figure 1 & 3.

17.Details of the interaction term (later referred to in subsequent sections) in the logistic regression models are missing. This is critical since the primary interest (according to the title of the paper) is effect modification necessitating interaction terms.

Response: Details of the interaction term was described in the Statistical analysis subsection in the Methods section: ‘The significance of modification effect by each covariate on periodontitis exposure-associated female breast cancer risk was examined by calculating the P-value of the coefficient associate with the product of each indicator of the covariate and the indicator of periodontitis using the Wald test.’ (page 9)

18.Perhaps the authors might consider making dataset and SAS code used available as SI

Response: Thank you for your comment. We have provided our dataset and SAS code in the supplementary files.

19.Results

Page 9 "After adjusting for age and index date"…this implies that age and index date were covariates in the regression model – this does not appear to be the case according to the statistical analysis section as it stands.

Response: We are sorry to make a wrong description. Actually, age and year of the index date are matching variables, not adjustment variables. We corrected this description as: ‘After adjusting for CCI and level of urbanization, periodontitis was significantly associated with the risk of female breast cancer (aOR, 1.12; 95% CI, 1.09–1.14).’ (page 9)

‘

20.It would be nice to have some supportive figures (e.g. partial effect plots) summarizing the statistical models.

Response: Thank you for your suggestion. We have conducted partial effect plots to summarize the statistical models and showed them in the supplementary files. (page 10)

21.Discussion

First paragraph talks about several studies one by one – it would be preferable to integrate this as continuous prose and convey the message that your work is supportive of these previous studies.

Response: We had revised our first paragraph of the Discussion section: ‘This is the first nationwide, population-based case control study which assessed the association between a history of periodontitis and the risk of female breast cancer development in Asia, and we have demonstrated a significant association of female breast cancer with periodontitis. The results remained robust in patients aged <65 years or ≥65 years and patients with a CCI = 0. Consistent with our finding, Soder et al. [20] selected 3,273 patients aged between 30-40 years and found that chronic periodontitis indicated by missing molars seemed to be associated with breast cancer. Of note, a meta-analysis study [21] found a periodontal disease significantly increased breast cancer risk. However, the risk of breast cancer development was not significantly increased among patients with periodontal disease and a history of periodontal therapy. Therefore, the author concluded a potential association between periodontitis and breast cancer. Freudenheim et al. [24] analyzed a cohort of 73,737 postmenopausal women and also revealed that periodontitis was associated with the risk of postmenopausal breast cancer. On the contrary, Farhat et al. [25] revealed no association between periodontal disease and overall breast cancer risk (HR 1.02, 95% CI; 0.94–1.10). However, in this study, no confounder was analyzed. In contrast, we analyzed the association between periodontitis and breast cancer adjusting for potential confounds, and we also found the important modification effects by age and comorbidities.’ (page 11)

22."In this study, one important finding is that the risk of breast cancer is modified by both age and comorbidities" - this statement is redundant since it is a repetition of the first sentence of the paragraph.

Response: We deleted this statement. (page 11)

23.Table 3 – It seems patient is the statistical unit as opposed to an adjustment variable as currently indicated at the foot of the table. Also urban should be "urbanization".

Response: We corrected it. (Table 3)

---

## [Decision Letter · Decision Letter 1]

4 Jul 2022

PONE-D-22-02756R1Effect Modification by Age and Comorbidities on the Association Between Periodontitis and Female Breast Cancer: A Population-based Case-control StudyPLOS ONE

Dear Dr. Chen,

Thank you for submitting your manuscript to PLOS ONE. After careful consideration, we feel that it has merit but does not fully meet PLOS ONE’s publication criteria as it currently stands. Therefore, we invite you to submit a revised version of the manuscript that addresses the points raised during the review process.

We look forward to receiving your revised manuscript.

Kind regards,

Antonis Valachis

Academic Editor

PLOS ONE

Journal Requirements:

Additional Editor Comments:

Although the authors have adequately respond to several of the reviewers' comments, there are some issues that need to be clarified before the manuscript be accepted for publication. Please, read the additional comments from the reviewers carefully and address all the remaining issues.

Reviewers' comments:

Reviewer's Responses to Questions

**Comments to the Author**

1. If the authors have adequately addressed your comments raised in a previous round of review and you feel that this manuscript is now acceptable for publication, you may indicate that here to bypass the “Comments to the Author” section, enter your conflict of interest statement in the “Confidential to Editor” section, and submit your "Accept" recommendation.

Reviewer #1: (No Response)

Reviewer #2: All comments have been addressed

2. Is the manuscript technically sound, and do the data support the conclusions?

Reviewer #1: Yes

Reviewer #2: Yes

3. Has the statistical analysis been performed appropriately and rigorously? 

Reviewer #1: Yes

Reviewer #2: Yes

4. Have the authors made all data underlying the findings in their manuscript fully available?

Reviewer #1: No

Reviewer #2: Yes

5. Is the manuscript presented in an intelligible fashion and written in standard English?

Reviewer #1: Yes

Reviewer #2: Yes

6. Review Comments to the Author

Reviewer #1: The authors have not addressed the comment on how their study was different from the previous study published on periodontal disease and cancer using the same Taiwan database (and also fail to mention this previous study) "A population-based study on the associations between chronic periodontitis and the risk of cancer" Shiu-Dong Chung et al.

Since breast cancer is a female cancer, stating that they studied females (vs both sexes) makes little sense as obviously prior studies on breast cancer risk were in women only. Please mention this study in the Introduction and this study should not be referred to as the "first" case-control on periodontal disease that is population based (as it is not).

This is not a retrospective case-control study. Retrospective case-control studies are studies that recruit cancer patients and ask about their past exposures; the exposure here is measured prior to the development of the disease. So, this is a retrospective cohort study, because the cohort exists and the analysis is conducted on a sample of the cohort retrospectively. Alternatively, you can call it a nested case-control study, which is nested in a cohort population.

Figure 1 – can you clarify what is meant by “unsured amount” given that half of the population is excluded for this reason (+ missing data) can you justify whether this was a reasonable exclusion and whether it might have caused some selection bias (in other words, how did the excluded population differ from the one included)?

Table 5. The title needs to be modified to clearly describe that the ORs presented are for the comparison of periodontal disease vs no periodontal disease stratified on age, CCI and urbanization. Many readers will misinterpret these results otherwise.

I don’t think the figures add any value, especially the ones that are based on categorial comparisons.

Please edit the abstract for grammar "assess that periodontitis" change to "assess whether periodontitis"

Reviewer #2: I enjoyed re-reviewing this interesting study and was pleased that the authors considered many of my suggestions. I only have two minor comments:

-Response to Reviewers - Reviewer #2: Point 12. The authors responded that "The variable of level of urbanization is an individual level variable based on individual’s residence region in the dataset." I believe this should be termed an ecological level variable since it describes properties of the region, and not each person.

Perhaps a very slight re-wording will suffice.

-The conclusion section looks a little peculiar with just one sentence. I would add a few more sentences or integrate into the discussion depending on the requirements of the journal.

Overall this is now essentially ready for publication and will be a valuable contribution to the literature.

7. PLOS authors have the option to publish the peer review history of their article (what does this mean?). If published, this will include your full peer review and any attached files.

Reviewer #1: No

Reviewer #2: No

---

## [Author Response · Author response to Decision Letter 1]

9 Jul 2022

Reviewer #1: 

1.The authors have not addressed the comment on how their study was different from the previous study published on periodontal disease and cancer using the same Taiwan database (and also fail to mention this previous study) "A population-based study on the associations between chronic periodontitis and the risk of cancer" Shiu-Dong Chung et al.

Response: Chung et al [20] observed an increased risk of a number of cancers among chronic periodontitis patients, and the adjusted hazard ratio was 1.23 for breast cancer. (page 4)

2.Since breast cancer is a female cancer, stating that they studied females (vs both sexes) makes little sense as obviously prior studies on breast cancer risk were in women only. Please mention this study in the Introduction and this study should not be referred to as the "first" case-control on periodontal disease that is population based (as it is not).

Response: We deleted ‘’first’’. (page 11)

3.This is not a retrospective case-control study. Retrospective case-control studies are studies that recruit cancer patients and ask about their past exposures; the exposure here is measured prior to the development of the disease. So, this is a retrospective cohort study, because the cohort exists and the analysis is conducted on a sample of the cohort retrospectively. Alternatively, you can call it a nested case-control study, which is nested in a cohort population.

Response: We rewrite as ‘’nested case-control study’’. (page 1, 2, 5,11,15)

4.Figure 1 – can you clarify what is meant by “unsured amount” given that half of the population is excluded for this reason (+ missing data) can you justify whether this was a reasonable exclusion and whether it might have caused some selection bias (in other words, how did the excluded population differ from the one included)?

Response: We are sorry to make a mistake here. Actually, we only excluded patients with missing data of residence region (a confounding factor in this study). Also, in 906,287 individuals from the 1997-2013 Longitudinal Health Insurance Database, we excluded excluded 458,873 men first and then excluded anther 26,126 women who had missing data of residence region. We have revised Figure 1. 

5.Table 5. The title needs to be modified to clearly describe that the ORs presented are for the comparison of periodontal disease vs no periodontal disease stratified on age, CCI and urbanization. Many readers will misinterpret these results otherwise.

Response: We have revised the tile as ‘Stratified Analysis of the Association Between Periodontitis and Female Breast Cancer Based on Age, Comorbidities and Level of Urbanization: A Population-based Nested Case-control Study.’

6.I don’t think the figures add any value, especially the ones that are based on categorial comparisons.

Response: Thank you for your suggestion. The other reviewer requested us to plot these figures based on both categorical and continuous comparisons. However, we have put them in the supplemental materials.

7.Please edit the abstract for grammar "assess that periodontitis" change to "assess whether periodontitis"

Response: We revised the sentence as: ‘ To conduct stratified analysis of the association between periodontitis exposure and the risk of female breast cancer based on age, comorbidities and level of urbanization.’ (page 2)

Reviewer #2: I enjoyed re-reviewing this interesting study and was pleased that the authors considered many of my suggestions. I only have two minor comments:

8.Response to Reviewers - Reviewer #2: Point 12. The authors responded that "The variable of level of urbanization is an individual level variable based on individual’s residence region in the dataset." I believe this should be termed an ecological level variable since it describes properties of the region, and not each person.

Perhaps a very slight re-wording will suffice.

Response: Thank you for your suggestion. We added a description regarding the variable of level of urbanization ‘The variable level of urbanization is an ecological level variable since it describes properties of the region.’ (page 8)

9.The conclusion section looks a little peculiar with just one sentence. I would add a few more sentences or integrate into the discussion depending on the requirements of the journal.

Response: We revised the conclusion section and integrated it into the Discussion section as: ‘In conclusion, this population-based nested case-control study revealed periodontitis exposure is significantly associated with the risk of breast cancer. We also found this association is modified by both age and comorbidities. Further studies are warranted to clarify the mechanisms.’(page 15)

---

## [Editor Report · Decision Letter 2]

12 Jul 2022

Stratified Analysis of the Association Between Periodontitis and Female Breast Cancer Based on Age, Comorbidities and Level of Urbanization: A Population-based Nested Case-control Study

PONE-D-22-02756R2

Dear Dr Chen,

We’re pleased to inform you that your manuscript has been judged scientifically suitable for publication and will be formally accepted for publication once it meets all outstanding technical requirements.

Kind regards,

Antonis Valachis

Academic Editor

PLOS ONE

---

## [Editor Report · Acceptance letter]

14 Jul 2022

PONE-D-22-02756R2 

Stratified Analysis of the Association Between Periodontitis and Female Breast Cancer Based on Age, Comorbidities and Level of Urbanization: A Population-based Nested Case-control Study 

Dear Dr. Chen:

I'm pleased to inform you that your manuscript has been deemed suitable for publication in PLOS ONE. Congratulations! Your manuscript is now with our production department. 

Kind regards, 

on behalf of

Assoc Prof Antonis Valachis 

Academic Editor

PLOS ONE